# Bacteriocin-Producing Lactic Acid Bacteria Strains with Antimicrobial Activity Screened from Bamei Pig Feces

**DOI:** 10.3390/foods11050709

**Published:** 2022-02-28

**Authors:** Jun Chen, Huili Pang, Lei Wang, Cunming Ma, Guofang Wu, Yuan Liu, Yifei Guan, Miao Zhang, Guangyong Qin, Zhongfang Tan

**Affiliations:** 1Henan Key Laboratory of Ion-Beam Bioengineering, School of Physics, Zhengzhou University, Zhengzhou 450052, China; chenjun6377@163.com (J.C.); liuyuan135254@163.com (Y.L.); 2School of Agricultural Sciences, Zhengzhou University, Zhengzhou 450001, China; pang@zzu.edu.cn (H.P.); guanyifei0904@163.com (Y.G.); miaozhang@zzu.edu.cn (M.Z.); qinguangyong@zzu.edu.cn (G.Q.); 3Academy of Animal Science and Veterinary Medicine, Qinghai University, Xining 800016, China; wanglei382369@163.com (L.W.); jim963252@163.com (G.W.); 4Bamei Pig Original Breeding Base of Huzhu County, Haidong 810600, China; macunming666@163.com

**Keywords:** lactic acid bacteria, screening, antimicrobial, bacteriocin, food industry

## Abstract

Lactic acid bacteria (LAB), which are characterized by producing various functional metabolites, including antioxidants, organic acids, and antimicrobial compounds, are widely used in the food industry to improve gut health and prevent the growth of spoilage microorganisms. With the continual incidence of foodborne disease and advocacy of consumers for gut health, LAB have been designated as vital biopreservative agents in recent years. Therefore, LAB with excellent antimicrobial properties and environmental tolerance should be explored further. In this study, we focus on screening the LAB strains from a specialty pig (Bamei pig) feces of the Tibetan plateau region and determine their antimicrobial properties and environmental tolerance to evaluate their potential probiotic values. A total of 116 LAB strains were isolated, from which the LAB strain Qinghai (QP)28-1 was identified as *Lactiplantibacillus* (L.) *plantarum* subsp. *plantarum* using 16S rDNA sequencing and *rec*A amplification, showing the best growth capacity, acid production capacities, environmental tolerance, hydrophobicity, antibiotic susceptibility, and bacteriocin production capacity. Furthermore, this strain inhibited the growth of multiple pathogens by producing organic acids and bacteriocin. These bacteriocin-encoding genes were identified using PCR amplification, including *plnS*, *plnN*, and *plnW*. In conclusion, bacteriocin-producing *L. plantarum* subsp. *plantarum* QP28-1 stands out among these 116 LAB strains, and was considered to be a promising strain used for LAB-related food fermentation. Moreover, this study provides a convenient, comprehensive, and shareable profile for screening of superior functional and bacteriocin-producing LAB strains, which can be used in the food industry.

## 1. Introduction

Probiotics are defined as “live microorganisms that, when administered in adequate amounts, confer a health benefit on the host.” [1,2,3]. As one of the mainly important probiotics, lactic acid bacteria (LAB) have been widely adopted in the food fermentation industry to improve the flavor and taste of products, including cheese, beer, and yogurt. They can also colonize the intestinal tract, promote the digestion and absorption of nutrients, and maintain the stability of the intestinal flora [4]. In recent years, LAB have been shown to produce antimicrobial substances, including organic acids, hydrogen peroxide, reuterin, acetoin, diacetyl, antifungal peptides, and bacteriocin [5,6,7,8], and LAB with broad-spectrum antibacterial properties are widely used for inhibiting spoilage bacteria and foodborne pathogenic bacteria and prolonging food shelf lives [9].

The antimicrobial activity of LAB has generally been attributed to two main modes, including the effect of organic acid and bacteriocin. Bacteriocins, an antimicrobial substance produced by bacteria, are a class of antimicrobial proteins or peptides synthesized on ribosomes, which can be degraded by proteases, are safe for the organism, and do not induce resistance. Bacteriocin has become a biological weapon against harmful food pathogens and has attracted interest as a tool for biopreservation, and it can be used as a natural antiseptic due to its bactericidal or bacteriostatic effects [10]. Antibiotics are the most widely used antibacterial substances, and their long-term use in large quantities can lead to the development of resistance in the organism; thus, bacteriocins are the most promising alternative to antibiotics as antibacterial substances. Likewise, bacteriocins can be applied as alternatives to existing preservatives in the food industry for the preservation of foods, such as dairy products, canned foods, and meat [11]. Many bacteriocin producers have been identified in LAB strains [12]. The first fully identified bacteriocin in LAB was nisin, synthesized by *Lactococcus lactis* [13]. Previous studies found that bacteriocins exert significant antimicrobial effects on many pathogens, including *Listeria monocytogenes*, *Bacillus cereus*, and *Staphylococcus aureus*, bringing valuable applications in the production of foods, such as yogurt, cheese, and sausages [14]. In recent years, there has been a lot of research on screening probiotics, especially LAB, and increasing numbers of LAB with excellent properties have been discovered and applied in fermented foods, food preservation, and intestinal health. However, in fact, there are still many difficulties to be solved and optimized in the process of screening probiotics and discovering bacteriocins, such as how to select more distinctive strains, how to screen target functional strains faster and more accurately, and how to discover bacteriocin-producing LAB efficiently.

Considering the above issues and according to our previous researches, we expect to obtain unique and excellent bacteriocin-producing LAB strains from samples obtained from the unique ecological region Tibetan plateau, which has specific geographical and climatic conditions, and has influenced the formation of specific microbiota in animals, fermented milk, vegetables, forage crops, and pasture [15,16]. In contrast to the rich natural resources, there are almost no studies on the isolation of bacteriocin-producing LAB from this region. Moreover, a well-integrated LAB probiotic candidate should have a variety of excellent characteristics, including rapid growth capacity, good environmental tolerance, outstanding antibacterial properties, and a good safety profile. Based on these factors, in this study, a series of experiments, including the combination of the traditional agarose diffusion approach with PCR bacteriocin-related gene detection, traditional analytical methods and PCA and heat map analysis, are designed and performed to screen the LAB strains for growth capacity, acid production capacity, acid–base tolerance, salt, bile salt, and the simulated gastrointestinal environment, antibiotic susceptibility, hemolysis, virulence genes, cell surface hydrophobicity, auto-aggregation and bacteriocin-producing genes to obtain bacteriocin-producing LAB by a rapid and accurate screening process as potential agents for microbial food spoilage prevention.

## 2. Materials and Methods

### 2.1. Sample Collection and Isolation of LAB Strains from Bamei Pigs

Feces were collected from a total of 44 pigs at different growth stages from the Huzhu County Pig Breeding Plant, which is situated at an altitude of over 3000 m in Qinghai Province, China. Samples were collected in sterile centrifuge tubes on 21 June 2020, and quickly transported to Zhengzhou in dry ice and stored in an ultra-low-temperature refrigerator at −80 °C. Feces samples (1 g) were added to 9 mL of sterilized water and then continuously diluted. The samples were 10-fold, 10^3^-fold, and 10^5^-fold diluted and inoculated on Man, Rogosa, and Sharpe (MRS) agar in an anaerobic incubator at 37 °C for 48 h. The creamy white raised colonies with different shapes and sizes were picked for isolation and purification. All these isolated strains were initially identified according to their physiological and biochemical characteristics, containing assays of Gram reactions, catalase activity, and gas production from glucose, as reported by Zhang et al. [17]. Only milky white, raised, catalase-negative, and Gram-positive isolates were selected for further assays.

### 2.2. Screening of the Strains with Broad-Spectrum Antibacterial Activity

In order to screen for LAB with antibacterial effects on a variety of pathogens, the antimicrobial activity of all the isolates from Bamei pig feces isolates was investigated by the Oxford cup double-layer plate method outlined by Muhammad et al. [5], using the pathogens *Escherichia coli* ATCC 30105; *Micrococcus luteus* ATCC 4698; *Staphylococcus aureus* ATCC 29213; *Pseudomonas aeruginosa* ATCC 27853; *Listeria monocytogenes* BAA; *Bacillus subtilis* ATCC 6633; *Salmonella enterica* subsp. *enterica* serovar Enteritidis ATCC 13076; and *Salmonella enterica* subsp. *enterica* serovar Typhimurium ATCC 43971 as indicator strains. Firstly, 20 mL of NA medium (NA, QingDao Hopebio Technology Co., Ltd., Qingdao, China) was placed in the plates as a lower layer of medium. Then, the 8 indicator bacteria that were cultured overnight in advance were injected into 5 mL of NA agar and cooled down naturally to about 50 °C with a 3% inoculum rate, mixed thoroughly, and immediately added to the plate to solidify them. Next, the sterilized Oxford cups were positioned on the surface of the NA agar and pressed gently. Finally, 200 µL of cell-free supernatant (CFS) of the isolated LAB was poured into the cups. After incubation at 37 °C for 24 h, the diameter of the inhibition zone was measured; this was repeated three times.

### 2.3. Identification of the Selected Strains by 16S rRNA Gene Sequence Analysis

The strains with significant antimicrobial activity in the initial screening were identified using prokaryotic 16S rRNA universal primers 27F (5′-AGAGTTTGATCCTGGCTCAG-3′) and 1492R (5′-GGTTACCTTGTTACGACTT-3′) [18], performed with the following PCR procedure: 3 min at 95 °C for pre-deformation; 35 cycles of 95 °C for 15 s, 60 °C for 15 s, and 72 °C for 90 s; and a final step of 72 °C for 5 min. The PCR products were sequenced by the Huada Biotech Company. The homologies between the gained sequences and those in GenBank were evaluated using BLAST analysis on the NCBI website (https://blast.ncbi.nlm.nih.gov/Blast.cgi, accessed on 16 August 2021). A bootstrap phylogenetic tree was generated based on the neighbor-joining method with MEGA 7 software (https://www.megasoftware.net/, accessed on 21 August 2021). Comparisons of partial amplification products of the *rec*A gene were utilized to differentiate strains of *Lactiplantibacillus* (L.) *plantarum* cluster, containing *L. pentosus*, *L. casei*, *L. paraplantarum*, *L. plantarum* subsp. *plantarum*, and *L. plantarum* subsp. *argentoratensis* [19]. A multiplex PCR assay was carried out using the *rec*A gene primers: paraF (5′-GTCACAGGCATTACGAAAAC-3′), pentF (5′-CAGTGGCGCGGTTGATATC-3′), planF (5′-CCGTTTATGCGGAACACCTA-3′), and pREV (5′-TCGGGATTACCAAACATCAC-3′). The PCR approach was adopted from the method reported by Wang et al. [20].

### 2.4. Physiological and Biochemical Characteristics and Tolerance

All the isolates were subjected to physiological and biochemical experiments to evaluate the multiple characteristics of the tested LAB strains, containing assays of the growth curve, acid production capacity, bile salt tolerance, and tolerance to low pH and extreme temperatures. In order to obtain LAB with excellent growth capacity and high resistance to the environment, a number of tolerance tests were performed on the strains obtained from the initial screening. The experimental methods and gradient settings for salt tolerance, extreme temperature tolerance, and acid–base tolerance of LAB strains are reported elsewhere by Zhang et al. [17]. To assess the alkali tolerance, LAB were inoculated with 1% inoculum in MRS liquid with different NaCl concentrations (3.5% and 6.5%) at 37 °C for 2 days; the growth of bacteria was observed or the absorbance at 600 nm was measured and compared with the growth of LAB in MRS liquid without NaCl as the control. Similarly, the temperature tolerance test was set at gradients of 5 °C, 10 °C, 30 °C, 45 °C, and 50 °C, and the acid tolerance test was established with pH gradients of 3, 3.5, 4, 4.5, 5, 8, 9, and 10.

Antibiotic susceptibility assays were conducted with the disk diffusion method, acquired from Anisimova and Yarullina [21], with minor modifications. In brief, 10 mL of MRS agar was poured into sterile plates and solidified. Then, 5 mL of the LAB suspension was mixed with MRS agar (50 °C) and poured into the upper layer of the plate, which then solidified. Next, 10 antibiotic discs (Oxoid Ltd., Basingstoke, U.K.), including ampicillin (AMP, 10 μg/disk), erythromycin (E, 15 μg/disk), colistin sulphate (CT, 10 μg/disk), penicillin (P, 10 μg/disk), ciprofloxacin (CIP, 5 μg/disk), gentamicin (CN, 10 μg/disk), chloramphenicol (C, 30 μg/disk), vancomycin (VA, 30 μg/disk), rifampicin (RD, 5 μg/disk), and tetracycline (TE, 30 μg/disk), were added to the surface of the plates. Finally, the diameters of the inhibition halos were measured after 24 h of incubation at 37 °C and interpreted as susceptible (S), moderately susceptible (MS), or resistant (R), based on the breakpoint values specified by the Clinical and Laboratory Standards Institute (CLSI, 2015).

The resistance properties under the simulated gastrointestinal tract (GIT) were assessed following the method of Zhang et al. [18]. For the simulated gastric juice, 0.35 g of pepsin was diluted in 0.2% saline and the pH was adjusted to 2.5 with 1 M HCL. For the simulated intestinal fluid, 0.1 g of trypsin, 1.8 g of bovine bile salt, 1.1 g NaHCO_3_, and 0.2 g NaCL were added to 100 mL sterile water, and the pH was adjusted to 8.0 with 1 M NaOH. The LAB cultured overnight were inoculated into the simulated gastric juice at 5% inoculum, vortex shaken for 30 s, and incubated at 37 °C for 3 h. After 3 h incubation at 37 °C, 100 μL of the 3 h solution was added to 900 μL of simulated intestinal fluid, which was then incubated at 37 °C for 30 s with vortex shaking. The LAB were incubated for 4 h at 37 °C. The viable counts of LAB after 0 h, 3 h, and 7 h were measured by the dilution spread method.

### 2.5. Cell Surface Hydrophobicity and the Auto-Aggregation Assay

The cell surface hydrophobicity and auto-aggregation ability of the screened LAB strains were investigated following the approach outlined by Sirichokchatchawan et al. [22]. Specifically, LAB strains cultured overnight in MRS broth were centrifuged, flushed three times with phosphate-buffered saline (PBS), and then resuspended in PBS until the OD was adjusted to 0.6 at 600 nm (A0). Subsequently, 3 mL of LAB cell suspension was added to 1 mL of xylene and allowed to stand for 40 min at 37 °C. The absorbance (At) of the aqueous phase at 600 nm was measured. The cell surface hydrophobicity (%) was assessed according to the following formula: hydrophobicity = (1 − At/A0) × 100%. The auto-aggregation was estimated as reported by De Souza et al. [23]: 1 mL of LAB suspension was allowed to stand for 2 h at 37 °C, and the absorbance (A1) of its upper liquid layer at 600 nm was determined. Auto-aggregation (%) = 1 − A1/A0 × 100%.

### 2.6. Screening of the LAB Strains Producing Broad-Spectrum Antibacterial Bacteriocin

The antimicrobial activity of the isolates was assessed by the agar diffusion assay [5], with *Escherichia coli* ATCC 30,105, *Micrococcus luteus* ATCC 4698, *Staphylococcus aureus* ATCC 29,213, *Pseudomonas aeruginosa* ATCC 27,853, *Listeria monocytogenes* BAA, *Bacillus subtilis* ATCC 6633, *Salmonella enterica* subsp. *enterica* serovar Enteritidis ATCC 13,076, and *Salmonella enterica* subsp. *enterica* serovar Typhimurium ATCC 43,971 as the indicator strains. In order to further explore the mechanism of antibacterial activity and to find out which substances are involved in the process of pathogen inhibition by LAB strains, a series of experiments were implemented on the effect of pH, catalase, and protease on the antibacterial ability. The CFS of LAB was adjusted to different pH values using 1 M HCl. The inhibitory ability of the CFS with different pH values against pathogens was determined by the above method. To inhibit the interference of hydrogen peroxide, 3% catalase was dissolved in PBS and added to the CFS at a concentration of 5.0 mg/mL of catalase, and the antibacterial activity of the CFS was detected. The untreated CFS was used as a control. Similarly, proteinase K, pepsin, and trypsin were added to the CFS at a final concentration of 1.0 mg/mL, noting the suitable temperature and pH of the different enzymes. The effect of proteinase K, pepsin, and trypsin on the antibacterial activity of the LAB strains was examined using the agar well diffusion method [24].

### 2.7. Amplification of Bacteriocin-Producing Genes of LAB

Bacteriocin-related gene PCR amplification experiments were performed on the strains, which exhibited antibacterial activity even after acid and hydrogen peroxide exclusion experiments, and were sensitive to protease. The involved genes encoding bacteriocin synthesis found from *L. plantarum* C11, V90, J23, J51, and WCFS1, and NC8 was identified from the NCBI database. The specific primers were designed (Table 1) for PCR amplification validation and the PCR amplification procedure was referenced in the report by Doulgeraki et al. [25]. The PCR products were detected by 1.5% agarose nucleic acid electrophoresis. The bands matching the expected fragment size were recovered by gum cutting and sent to the Huada Biotech Company Co., Ltd. (Zhengzhou, China); the results were compared with the sequences of the genes encoding these bacteriocin-related genes in the NCBI database.

### 2.8. Statistical Analysis

Principal component analysis (PCA) was performed in Excel combined with the XLSTAT program. Heat map analysis was carried out in ClustVis (https://biit.cs.ut.ee/clustvis, accessed on 17 September 2021). Statistical analysis was conducted using the SPSS 13.0 software (SPSS, Inc., Chicago, IL, USA). One-way ANOVA tests were applied to analyze the significance of differences (*p* < 0.05).

## 3. Results

### 3.1. Screening of the Broad-Spectrum Antibacterial Active LAB

A total of 116 presumptive LAB isolates was obtained from the fecal samples of 44 Bamei pigs from the original breeding farm of Bamei pigs in Huzhu County, Qinghai Province, China. Eight LAB strains stood out from all the isolates because of their excellent antibacterial activity, which inhibited at least three pathogens (Appendix A). By combining 16S rRNA gene sequence analysis with *rec*A gene multiple detection, strains QP3-2, QP19-1, and QP28-1 were identified as *L. plantarum* subsp. *plantarum*, and QP4-2 and QP33-2 could be assigned to *L. casei* (Appendix A). In the subsequent screening process, *Lactococcus lactis* QP20-2 and *Enterococcus hirae* QP22-2 were omitted due to their weak ability to survive in high-salt, high-temperature, or low-pH environments and their low acid-production capacity (Appendix A). *L. casei* QP4-2 and *Enterococcus hirae* QP23-1 were also excluded because of their lower survival in GIT, presented in Appendix A.

### 3.2. Preliminary Validation of the Antimicrobial Substance Produced by LAB

In order to investigate the mechanism of bacterial inhibition of these LAB and whether the acid or the bacteriocin caused the inhibition, some experiments were performed, including assessing the influence of pH, protease, and hydrogen peroxide on the antimicrobial activity. As shown in Figure 1, the antibacterial potential of LAB against all pathogens decreases with elevated pH values, which suggests that the impact of pH on the antimicrobial activity is evident. When the pH of the CFS was 5.0, all four strains showed broad-spectrum inhibition against six of the pathogenic bacteria, with QP28-1 reaching a maximum inhibition circle size of 23.24 mm against *M*. *luteus* ATCC 4698. Compared with QP3-2 and 19-1, LAB strains QP28-1 and 33-2 still displayed an inhibition of *Staph*. *aeruginosa* ATCC 29213. However, when the pH increased to 5.5, the inhibitory ability of these strains showed a significant decrease; notably, the inhibition of *L. monocytogenes* BAA completely disappeared, and the original inhibition of *Staph*. *aeruginosa* ATCC 29213 by QP28-1 and QP33-2 also disappeared. All four strains still retained a high inhibitory activity against *E*. *coli* ATCC 30105, *Ps*. *aeruginosa* ATCC 27853, and *B*. *subtilis* ATCC 6633, and did not lose performance with the increase in pH, which suggested that these LAB strains of QP3-2, QP28-1, and QP33-2 were good candidates for antimicrobial activity.

Considering that LAB may produce hydrogen peroxide (H_2_O_2_) to inhibit pathogenic bacteria, the bacteriostatic ability of CFS treated by catalase was examined, as shown in Table 2. The inhibitory ability of most of the CFS was maintained after catalase treatment, although it was also noted that the inhibitory ability of four LAB strains against *Staph*. *aeruginosa* ATCC 29213 and *Salmonella enterica* subsp. *enterica* serovar Enteritidis ATCC 13076 disappeared. Nevertheless, when the CFSs of these four strains were treated with proteinase K, pepsinum, and tryptase, almost all of the inhibitory effects completely disappeared, which implied that the antibacterial substance was sensitive to proteases; thus, it was tentatively determined to be a protein.

### 3.3. Selection of Potentially Probiotic Isolates

In order to assess the similarity and variability between the LAB strains in various probiotic phenotypes, the probiotic characteristics, including bile salt tolerance, gastrointestinal tract (GIT), hydrophobicity, auto-aggregation, and antimicrobial activity of the isolated LAB strains were analyzed through multivariate analysis based on PCA. As illustrated in Figure 2, the first (PC1) and the second (PC2) principal components represented 81.575% and 11.099% of all the variables (bile salt tolerance, GIT, hydrophobicity, auto-aggregation, and antimicrobial activity), indicating significant differences in the probiotic characteristics of the LAB strains. The eight LAB strains were divided in three areas based on PCA. Five of the LAB strains containing QP3-2, QP4-2, QP19-1, QP28-1, and QP33-2 were clustered close to each other, which showed similar probiotic characteristics and exhibited good tolerance and antimicrobial activity. In contrast, QP20-2, QP23-1, and QP22-2 were listed separately, which displayed weak tolerance and inhibition to pathogenic bacteria in the above study results. Furthermore, a heat map was generated to cluster probiotic strains based on their phenotype (bile salt tolerance, GIT, hydrophobicity, auto-aggregation, and antimicrobial activity). As shown in Figure 3, the probiotic phenotype heat map clustered eight LAB strains into two clusters and four sub clusters, A, B, C, and D. Overall, the probiotic properties of cluster A and B and QP33-2 from cluster D were better than the other clusters. The strain QP4-2 from cluster B was excluded because of its low GIT value, which means that it is difficult to survive and colonize the gastrointestinal tract. The strains QP23-1, QP20-2, and QP22-2 were not selected because they displayed lower values for bile salt tolerance, hydrophobicity, auto-aggregation, and antimicrobial activity. The LAB strains QP19-1, QP28-1, QP3-2, and QP33-2 were selected as potentially probiotic isolates because they exhibited excellent probiotic phenotypes, especially in antimicrobial activity.

### 3.4. The Discovery of Bacteriocin-Producing LAB

The analysis of the above results speculated that genes that can encode bacteriocin may be present in these LAB strains; therefore, some of the most common genes in the LAB that encoded bacteriocin were screened and tested. Using the total DNA of the 4 screened strains as templates, PCR amplification was performed using 13 pairs of specific primers. Remarkably, strain QP28-1 was found to possess three genes that could encode bacteriocin. As shown in Figure 4, *plnS*, *plnN,* and *plnW* from QP28-1 all amplified a single band and matched the expected fragment size, which suggested that the LAB strain of QP28-1 could encode these corresponding bacteriocins.

## 4. Discussion

This study focused on screening for LAB superior performance; how to screen for the desired LAB based on the assessments of desired probiotic attributes is a rigorous scientific process. According to the criteria given by the FAO/WHO [1,2,3], the screening of probiotic bacteria should meet the following three basic characteristics: a tolerance of the selective environment of gastrointestinal mucosa; an adherence to the intestinal cells of the host; and the secretion or production of antibacterial substances as breakdown products [26]. The possible mechanism of this probiotic effect is that LAB secrete and produce antimicrobial substances, such as lactic acid, acetic acid, and bacteriocin, which prevent the proliferation of coliforms and other pathogens [27]. On the one hand, LAB strains produce organic acid and lower the pH of the environment through fermentation, making it difficult for pathogenic bacteria to tolerate acidic environments and survive [28]. For example, Cervantes-Elizarrarás et al. found that the organic acid produced by LAB can suppress some Gram-negative pathogens by permeating the cell membrane, thus acidifying the cytoplasm, interrupting its function, and destroying acid-sensitive enzymes [29]. On the other hand, some LAB strains produce bacteriocin, which is a kind of polypeptide substance used to block the growth of some foodborne pathogens and deleterious bacteria, such as *Escherichia coli*, *Staphylococcus*, and *Salmonella* [30]. In our study, the LAB strains isolated and screened from the intestines of local breeding Bamei pigs from the alpine and high-altitude Qinghai–Tibetan Plateau were analyzed for antimicrobial activity and spectrum, and the bacteriocin produced by LAB QP28-1 was detected and identified. The results show that the eight strains of LAB with the best inhibition effects on a variety of pathogens include *M*. *luteus* ATCC 4698, *P*. *aeruginosa* ATCC 27853, and *B*. *subtilis* ATCC 6633, and also have a significant inhibition effect on pathogens *E*. *coli* ATCC 30105, *S*. *aureus* ATCC29213, *L. monocytogenes* BAA, and *Salmonella enterica* subsp. *enterica* serovar Typhimurium ATCC 43971, indicating their promising application prospects. In previous studies on screening LAB, most of them used one or two pathogenic strains as indicator bacteria, especially *Salmonella* and *E*. *coli*, which are prone to cause diarrhea [31]. In comparison, we used eight kinds of pathogenic bacteria as indicator bacteria and screened LAB with a broad-spectrum antibacterial activity for multiple pathogenic bacteria, which improved the application value. In terms of the isolation source, we isolated the LAB from the intestine of pigs, which means that our strain possessed higher tolerance to acid, bile salts, and the gastrointestinal environment; therefore, it exhibited a higher chance of survival and colonization in potential agricultural production, as well as in the development of feed additive applications. In the process of screening LAB with good bacterial inhibition, it is important to consider not only the level of unilateral inhibition performance, but also some other properties, including high growth capacity, tolerance, colonization capacity, and safety. A higher strain multiplication capacity ensures that more strains grow and multiply, improving the overall antibacterial inhibition effect. The low survival rate in GIT (Appendix A) and bile salt environments (Appendix A) implied that strains QP4-2 and QP23-1 have difficulty surviving in the host, which runs counter to the probiotic definition. Cell surface properties, such as hydrophobicity and aggregation, are essential indicators of probiotic adherence to human intestinal epithelial cells [32,33]; previous studies illustrated that the auto-aggregation of cells increases with the colonization ability [34]. The screened strains were also tested and *L. casei* QP33-2 exhibited significantly higher hydrophobicity and self-aggregation than other strains (Appendix A). Furthermore, the principal component analysis (PCA) presented in Figure 2 suggests that there are differences in the probiotic attributes of the screened LAB. However, some of the LAB strains virtually displayed a similarity in the bile salt tolerance, gastrointestinal tract (GIT), hydrophobicity, auto-aggregation, and antimicrobial phenotypes of probiotics. Previously, researchers also used PCA to screen for probiotic candidate, investigating the similarities and differences in the probiotic properties of obtained strains [35]. The heat map in Figure 3 clustered the LAB strains based on their phenotypic properties, and all LAB strains were classified into four subtypes. *L. plantarum* subsp. *plantarum* QP19-1 and QP28-1 from cluster A were screened due to their significantly better performance than the other strains. The utility of PCA and heat-map analysis in the screening of probiotic isolates with different phenotypes was demonstrated in a previous study [36], which provided sufficient tools in the process of seeking candidate probiotic strains. The hemolytic activity and common virulence, including the factor genes and biogenic amine genes of the eight screened LAB strains, were tested, and no hemolytic activity or virulence gene was found, thus proving that these LAB strains are safe.

In addition to the acid and hydrogen peroxide produced by metabolism of LAB, bacteriocin has important application prospects as a safe and effective bactericidal substance. The agar diffusion method is the most traditional means of validation for identifying bacteriocin-producing strains, but due to the low yield of bacteriocin, the inhibition effect is not significant after the exclusion of acid and hydrogen peroxide, and the accuracy and reproducibility are poor. In recent years, several methods have been established to screen bacteriocin-producing LAB as food preservatives for the food industry. With the ongoing accumulation of biological information, such as encoding genes, the PCR rapid detection method improved the efficiency and reliability in screening bacteriocin-producing strains [37]. Macwana et al. [38] designed 42 pairs of primers through the reported structural genes of different species of lactic acid bacteriocins, and identified 3 novel bacteriocin-producing LAB, *Lactococcus lactis* RP1, *Pediococcus acidilactici* Bac3, and *Latilactobacillus sakei* JD1, through real-time fluorescence quantitative PCR using the DNA of the target strains as templates and then comparing the sequences [38]. Więckowicz et al. [39] categorized the reported bacteriocin sequences into 5 major classes according to homology and designed multiple pairs of specific primers based on the structural characteristics to identify 40 bacteriocin genes encoding class IIa bacteriocins in the genome from cheese. The PCR amplification method was demonstrated to enable high-throughput screening and analysis in samples producing specific bacteriocin. In this study, the CFS of LAB strain QP28-1 retained its antibacterial activity after the successive removal of acids and hydrogen peroxide, indicating that *L. plantarum* QP28-1 could produce bacteriocin. The PCR analysis was carried out to identify the bacteriocin genes of QP28-1 by using 13 pairs of primers designed to be specific for the genes related to the biosynthesis of bacteriocin. The structural gene *plnN/S/W* in the genome of strain QP28-1 was successfully amplified by PCR, and was found to be identical to the structural gene *plnN/S/W* encoding class IIb bacteriocins in *L. plantarum* C11 after comparison [40].

Strain QP28-1 was identified as a broad-spectrum, highly efficient bacteriostatic *L. plantarum* strain using the PCR method, which produced class IIb bipeptide bacteriocin. The rapid identification of bacteriocin-related genes using the PCR method accelerated the screening speed and improved the accuracy. However, the rapid detection method also has some limitations, because the PCR method only draws comparisons with the reported bacteriocin genes, resulting in many novel bacteriocins not being detected; therefore, there is still much room for improvements to the method. As such, this experiment was validated by combining the traditional agar diffusion method with the PCR method for bacteriocin production. Further studies on the synthesis and mechanism of action of the *L. plantarum* QP28-1 bacteriocin gene and the improvement in bacteriocin production are being carried out in our laboratory. Many bacteriocins were found to have tremendous potential as food pretreatment agents in food spoilage preservation [41]. In practice, bacteriocins used as additive preservatives capable of inhibiting pathogens are often implemented as inoculated probiotic themselves or as purified bacteriocins [42]. It is the antimicrobial activity and non-toxic nature of bacteriocins as potential inhibitors of various pathogens that inspired the interest of researchers in bacteriocins [43]. The results show that *L. plantarum* QP28-1 exhibits a good spectrum of activity towards multiple pathogens and can produce bacteriocin, indicating that it can be used for preservation and inhibiting microbial food spoilage.

## 5. Conclusions

In this study, eight LAB strains were isolated from the feces of Bamei pigs with excellent broad-spectrum antibacterial activity. *L. plantarum* subsp. *plantarum* QP3-2, QP19-1, QP28-1, and *L. casei* QP4-2, QP33-2 showed broad antibacterial activity against the pathogenic bacteria *E*. *coli* ATCC 30105, *M*. *luteus* ATCC 4698, *S*. *aureus* ATCC 29213, and *Salmonella enterica* subsp. *enterica* serovar Enteritidis ATCC 13076. Combining the accurate screening processes and assessment methods of PCA and heat-map analysis, four strains of LAB, including *L. plantarum* subsp. *Plantarum* QP3-2, QP19-1, QP28-1, and *L. casei* QP33-2, were selected for the further study of the inhibition mechanism. Moreover, these four LAB strains still exhibited antimicrobial activity after acid exclusion and hydrogen peroxide exclusion, suggesting the presence of bacteriocins. The PCR products of QP28-1 bacteriocin-related genes revealed that QP28-1 could produce bacteriocin. In conclusion, *L. plantarum* subsp. *plantarum* QP28-1 exhibited a high potential application value for preventing microbial food spoilage and agricultural applications, which produce bacteriocin. These screened LAB strains could be developed as functional antimicrobial agents in the future, thus warranting a more detailed and in-depth exploration.

## Figures and Tables

**Figure 1 foods-11-00709-f001:**
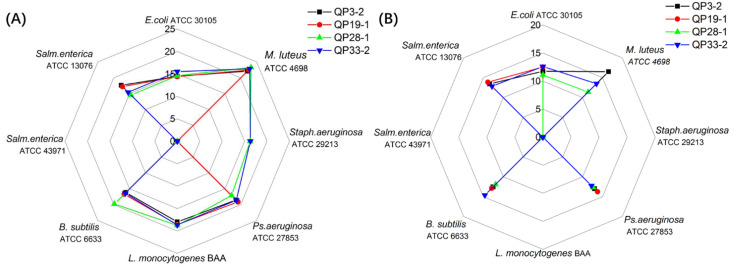
The influence of pH on antimicrobial activity. The value represents the diameter of the inhibition circle, containing the diameter of the Oxford cup (10.00 mm). (**A**) Antimicrobial activity of LAB strains at pH = 5; (**B**) antimicrobial activity of LAB strains at pH = 5.5.

**Figure 2 foods-11-00709-f002:**
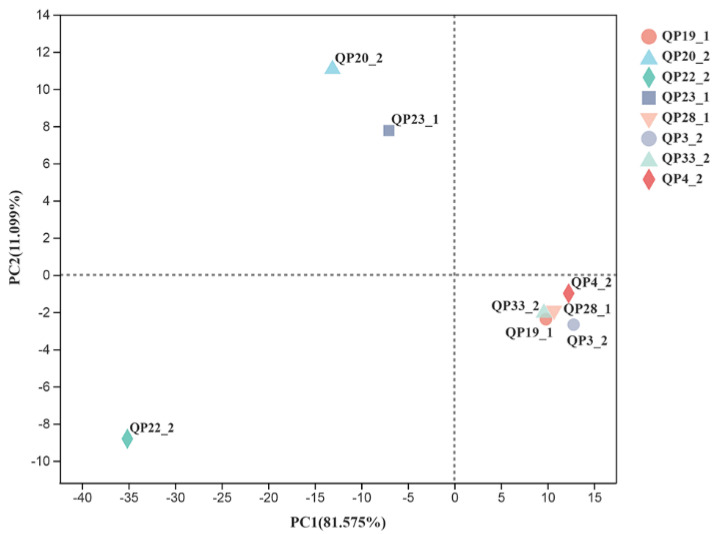
Principal component analysis (PCA) of LAB phenotypes based on the bile salt tolerance, gastrointestinal tract (GIT), hydrophobicity, auto−aggregation, and antimicrobial activity.

**Figure 3 foods-11-00709-f003:**
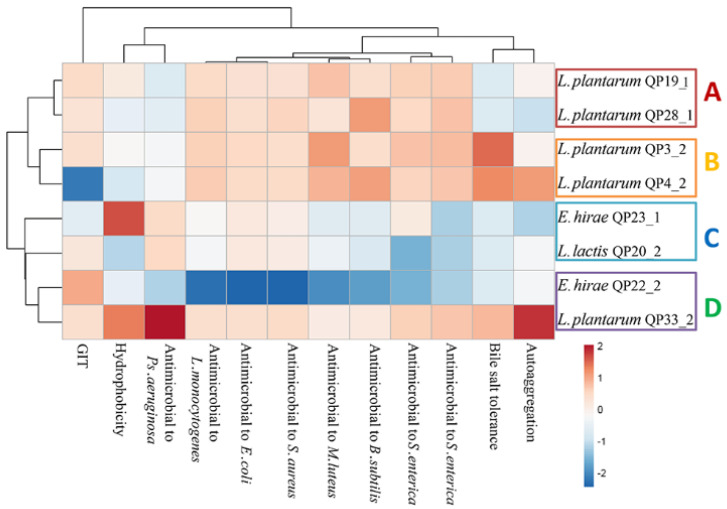
The heat map of LAB phenotypes, including bile salt tolerance, gastrointestinal tract (GIT), hydrophobicity, auto-aggregation, and antimicrobial to *E*. *coli*, *M*. *luteus*, *S*. *aureus*, *Ps*. *Aeruginosa*, *L. monocytogenes*, *B*. *subtilis*, and *S*. *enterica*.

**Figure 4 foods-11-00709-f004:**
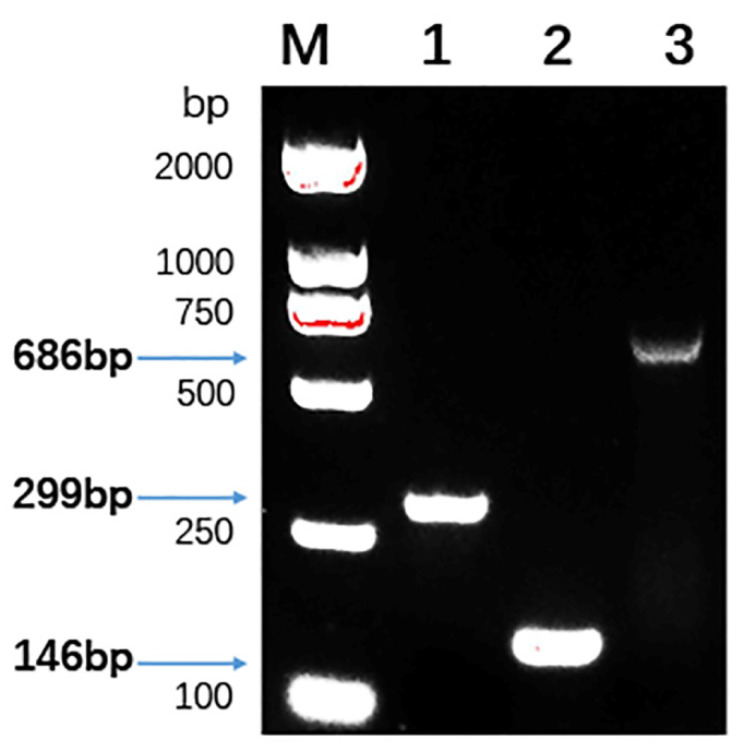
Agarose gel electrophoresis of the PCR products of QP28-1 bacteriocin-related genes. Lane M: DNA marker; lane 1: the product of *plnS* (299 bp); lane 2: the product of *plnN* (146 bp); and lane 3: the product of *plnW* (686 bp).

**Table 1 foods-11-00709-t001:** PCR primers for LAB bacteriocin-producing genes.

Target Gene	Primer Sequence (5′ to 3′)		PCR Product Size/bp	Annealing Temperature/°C
	Forward	Reverse		
*nis*	GGCATAGTTAAAATTCCCCCC	CAGGTTGCCGCAAAAAAAG	428	53
*ent*	GGGTACCACTCATAGTGGAA	CCAGCAGTTCTTCCAATTTCA	412	53
*plnJK*	GCCACAAAGAGCACTAACA	CATACAAGGGGGATTATTT	427	54.8
*plnS*	ATGGCACACTCAAATAAAC	TCAACAATAATGAGCACGA	299	58.3
*plnA*	ATGAAAATTCAAATTAAAGG	TTACCATCCCCATTTTTTA	146	55
*plnXY*	ATTCAGCGATTAGCATTG	GGAGCCATAAACTCTTCTT	286	52.9
*plNC8IF*	TTGGCGGAAAAACAAAGACT	TCAGCATGTCATTTCACCATC	114	52.5
*plnW*	ATGTTACAGAAGAATTTACGGT	TTAGCTAGGAACCAACCAG	686	54.8
*pln423*	TATGATGAAAAAAATTGAAAAAT	CCAAAGATAATCCCCCCCCAT	197	50
*plnN*	ATTGCCGGGTTAGGTATCG	CCTAAACCATGCCATGCAC	146	51.9
*ped*	GGTAAGGCTACCACTTGCAT	CTACTAACGCTTGGCTGGCA	332	53
*plnEF*	TGATGGCTTGAACTATCCGTG	CATACAAGGGGGATTATTT	385	58.3
*plnQ*	TGAAATCCTACAATATGAAATTGAACCGCGA	TTATTTTCTCTTACTTGTAAAGGCTCTCAA	188	55

**Table 2 foods-11-00709-t002:** The antibacterial activity of LAB strains after protease treatments.

Strain	Treatment	*E. coli* ATCC 30105	*M. luteus* ATCC 4698	*Staph. aureus* ATCC 29213	*Ps. aeruginosa* ATCC 27853	*L. monocytogenes* BAA	*B. subtilis* ATCC 6633	*Salm. Enterica* ATCC 43971	*Salm. Enterica* ATCC 13076
QP 3-2	catalase	13.47	19.21	0.00	15.99	0.00	15.61	0	0
	proteinase K	0	0	0	0	0	11.86	0	0
	pepsinum	0	0	0	0	0	0.00	0	0
	tryptase	0	0	0	0	0	0.00	0	0
QP 19-1	catalase	13.13	20.25	0	17.73	18.14	17.36	12.19	0
	proteinase K	0	0	0	0	0	11.94	0	0
	pepsinum	0	0	0	0	0	0	0	0
	tryptase	0	0	0	0	0	13.32	0	0
QP 28-1	catalase	12.21	16.11	0	15.94	0	14.86	0	0
	proteinase K	0	0	0	0	0	0	0	0
	pepsinum	0	0	0	0	0	0	0	0
	tryptase	0	0	0	0	24.20	0	0	0
QP 33-2	catalase	12.80	16.29	0	16.33	0.00	15.77	0	0
	proteinase K	0	0	0	0	0	0	0	0
	pepsinum	0	0	0	0	0	0	0	0
	tryptase	0	0	0	0	0	0	0	0

Note: the values represent the diameter (mm) of the bacterial inhibition circle (*n* = 3). The inhibition zone contains the external diameter of the cup (10.00 mm).

## Data Availability

The datasets presented in this study can be found in online repositories. The 16S rRNA gene sequence of LAB strains QP3-2, QP4-2, QP19-1, QP28-1, QP20-2, QP22-2, QP23-1, and QP33-2 used to support the findings of this study were deposited in the GenBank repository with accession numbers OM049400, OM049401, OM049402, OM049403, OM049404, OM049405, OM049406, and OM049407, respectively. http://www.ncbi.nlm.nih.gov, accessed on 29 December 2021.

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
