# Peer review of "Bacteriocin-Producing Lactic Acid Bacteria Strains with Antimicrobial Activity Screened from Bamei Pig Feces"

_foods, 2022, doi:10.3390/foods11050709_

Round 1

Reviewer 1 Report

Dear Authors,

I agree with you that screening for new lactic acid bacteria isolates is an important issue. LAB cells are beneficial to human health and our knowledge on antibacterial and antigungal properties of individual strains is still insufficient. Nevertheless, I cannot agree that the manuscript which I have reviewed, is innovative enough. I found not much elements which can be considered as scientific report. I do agree that Authors prepared a comprehensive and shareable profile for screening of bacteriocin-producing probiotic LAB.

The main disadvantages of the paper are:

  • Introduction - it provides many obvious information which are known by the scientists dealing with LAB characteristics. I do not agree that fewer study are related to the inhibition of pathogenic bacteria by baceriocin produced by LAB. There is a plenty of such studies. I also do not understand why a whole paragrapgh is related to Bamei pig. The choice of animal for strain isolation should be mentioned only in materials and methods section;
  • the aim of the study - I would recommend to re-write the manuscript and to change the aim of the study, in my opinion the only scientific aim was to obtain bacteriocin producing LAB by accurate screening process. I would also encourage you to underline why it was important from the practical point of view and so to change the title;
  • methods - in paragraph 2.4 please put more details on methods used which are described by Zhang et al.;
  • results - I would recommend to delete paragraphs from 3.1 to 3.5 and put it into Supplementary Material, describe in materials section why 8 strains have been chosen for experiments;
  • Discussion - this section is too long. The first paragraph should be removed, because it provides the same information which are written in introduction section. Put more emphasis on your own achievements connected with statistical data analysis. 
  • Conclusions are not much informative.

Reviewer 2 Report

General comments

Unfortunately, the manuscript is badly written - there is confusion. Lack of consistency and logic in the text. It is not known what the causes of the methods are poorly described. A poorly scientific language. Some terms are used interchangeably without instantiation.

Why Bamei pigs (Sus scrofa) were selected as model for isolation of LAB? According to FAO/WHO probiotic LAB for humans should originate from human…

The English language must be improved to more scientific and the style must be corrected.

Detailed comments

  • Authors should apply new nomenclature of Lactobacillus in the whole manuscript, see: http://lactotax.embl.de/wuyts/lactotax/
  • Introduction: give the latest definition of “probiotics”.
  • Line 29: “…by producing organic acids lactic…” correct “organic acids lactic”.
  • Line 54: give space in “Escherichiacoli”.
  • Lines 47-55 should be transferred to Discussion.
  • Line 53 -Authors start to write about bacteriocins. The same starts from line 70.
  • Introduction must be more coherent and structured. In this form it is chaotic. The aim should be clearly worded and separate from the rest of the information and without digressions.
  • How long the feces were stored at -80oC?
  • Line 119- why in 30oC? Why not in 37oC?
  • There are many typing errors, which should be corrected.
  • Line 131: Salmonella enterica or Typhimurium?
  • Lines 120-124 - are the same information in lines 159-160?
  • Paragraph 2.4 is chaotic.
  • Table 2: there are typing errors such as different font sizes(?), unnecessary capital letter etc. The caption says nothing.
  • I can see results for 8 strains. What about the rest? Why these 8? Emphasise please the choice.
  • Paragraph 3.3 – what tolerance? To what? pH? Bile salts? Together or separately? Physiological, biochemical characteristics of what?
  • Figure 3 caption. There is a mess. It must be improved.
  • Figure 3 caption: What simulated gastrointestinal conditions? pH? Bile salts? Together or separately?
  • Table 6: what was the positive control (ie chemical substance) for the test?
  • The Antimicrobial susceptibility testing should be teste according to EUCAST. Look at: https://www.eucast.org/ast_of_bacteria/disk_diffusion_methodology/ What is Authors method? It is not clearly presented.
  • Paragraph 3.7. Authors should use a term “potentially probiotic” in this case, not probiotic. Probiotic the strains could be after in vivo experiments.

Author Response

Please see the acttachment

Round 2

Reviewer 1 Report

Dear Authors,

I do appreciate your efforts in improving your manuscript. Thank you for taking my suggestions into consideration. Wish you good luck in further experiments.

with regards

Reviewer 2 Report

Point 5: Introduction: give the latest definition of “probiotics”.

Response 5: A latest definition of “probiotics” has been added to the manuscript. Please see P2, L55-L57.

[1] Morelli, L.; Capurso, L. FAO/WHO Guidelines on probiotics 10 Years later foreword. J Clin Gastroenterol 2012, 46, S1-S2, doi:DOI 10.1097/MCG.0b013e318269fdd5.

I thought that this definition is the latest: https://pubmed.ncbi.nlm.nih.gov/28611480/

Point 14: Line 131: Salmonella enterica or Typhimurium?

Response 14: Both salmonellae used as indicator bacteria in the experiments were Salmonella enterica. Salmonella typhimurium was not used in our experiments.

I thought that the correct name of Salmonella enterica is Salmonella enterica ssp. enterica var. Typhimurium? Follow: https://www.ncbi.nlm.nih.gov/Taxonomy/Browser/wwwtax.cgi?mode=Tree&id=90371&lvl=3&p=has_linkout&p=blast_url&p=genome_blast&lin=f&keep=1&srchmode=1&unlock
